

# Is it feasible to estimate radiosonde biases from interlaced measurements?

Stefanie Kremser[1], Jordis S. Tradowsky[1,2,3], Henning W. Rust[2], and Greg E. Bodeker[1]

[1]Bodeker Scientific, 42 Russell Street, Alexandra, New Zealand
[2]Freie Universität Berlin, Institute for Meteorology, Carl-Heinrich-Becker Weg 6-10, Berlin
[3]National Institute of Water and Atmospheric Research, Lauder, New Zealand

*Correspondence to:* Stefanie Kremser (stefanie@bodekerscientific.com)

**Abstract.** Upper-air measurements of essential climate variables (ECVs), such as temperature, are crucial for climate monitoring and climate change detection. Because of the internal variability of the climate system, many decades of measurements are typically required to robustly detect any trend in the climate data record. It is imperative for the records to be temporally homogeneous over many decades to confidently estimate any trend. Historically, records of upper-air measurements were pri-
marily made for short-term weather forecasts and, as such, are seldom suitable for studying long-term climate change as they lack the required continuity and homogeneity. Recognizing this, the Global Climate Observing System (GCOS) Reference Upper-Air Network (GRUAN) has been established to provide reference-quality measurements of climate variables, such as temperature, pressure and humidity, together with well characterized and traceable estimates of the measurement uncertainty. To ensure that GRUAN data products are suitable to detect climate change, a scientifically robust instrument replacement strat-
egy must always be adopted whenever there is a change in instrumentation. By fully characterizing any systematic differences between the old and new measurement system a temporally homogeneous data series can be created. One strategy is to operate both the old and new instruments in tandem for some overlap period to characterize any inter-instrument biases. However, this strategy can be prohibitively expensive measurement sites operated by national weather services or research institutes. An alternative strategy that has been proposed is to alternate between the old and new instruments, so-called interlacing, and then
statistically derive the systematic biases between the two instruments. Here we investigate the feasibility of such an approach specifically for radiosondes, i.e. flying the old and new instruments on alternating days. Synthetic data sets are used to explore the applicability of this statistical approach to radiosonde change management.

## 1  Introduction

Radiosondes are indispensable for monitoring the upper-air as they provide high vertical resolution in situ observations of
temperature, pressure and water vapour between the surface and the upper troposphere/lower stratosphere. Determining long-term temperature trends from radiosonde measurements is challenging because changes in instrumentation can, among other things, introduce discontinuities in the measurement time series (see Fig. 1). Since radiosonde measurements are primarily made to provide the data needed to constrain weather forecasts and not to detect long-term changes in climate, little attention has been paid to ensuring the long-term homogeneity of the measurement record when changing from one instrument to another.



As a result, radiosonde data records typically fall short of the standard required to reliably detect changes in climate. Another cause of inhomogeneities in the record is undocumented changes in data processing (Thorne et al., 2011b). While much effort has been spent attempting to remove discontinuities in radiosonde data records (e.g. Sherwood et al., 2005; Randel and Wu, 2006; Haimberger et al., 2012), lack of confidence in the long-term homogeneity erodes confidence in derived trends. Seidel

and Free (2006) used upper-air temperatures from the NCEP-NCAR reanalysis (Saha et al., 2010) to investigate the effects of sampling frequency, changes in observation schedule, and the introduction of inhomogeneities, to the radiosonde climate data record. Their results indicate that introducing inhomogeneities into a temperature time series provides the most significant source of uncertainty on trend estimates. Maintaining the temperature measurement stability to within 0.1 K for periods of 20 to 50 years, avoids uncertainties in trend estimates in at least 99% of cases (Seidel and Free, 2006). With a weaker stability

requirement of 0.25 K, the uncertainty on a 50 year trend estimate increases by about 5% for twice-daily sampling. Rust et al. (2008) showed that inhomogeneities in temperature measurements can cause spurious memory, leading to larger uncertainty for statistics derived from these series. The results of these studies demonstrate the need to account for any inhomogeneities in the measurement time series prior to any trend analysis. The GCOS (Global Climate Observing System) Reference Upper-Air Network (GRUAN) was established to provide reference-quality measurements of atmospheric ECVs, suitable for reliably

detecting changes in global and regional climate on decadal scales. To avoid compromising the integrity of the long-term climate record, it is essential that any change, e.g. in the instrumentation or data processing, is adequately assessed before the change is implemented. For example, when transitioning from one radiosonde type to another, inter-comparison between both radiosonde types is required to assess a potential systematic difference between the radiosondes and to correct for it, ensuring a continuous homogeneous data set without any introduced discontinuities. Typically, intercomparisons of measurements from

dual or quadruple (two of each instrument-type) radiosonde flights are used to robustly detect systematic differences between the instruments (e.g. Luers and Eskridge, 1998; Steinbrecht et al., 2008; Jensen et al., 2016). Results presented in Steinbrecht et al. (2008) indicated that temperature biases often increase significantly with increasing altitude, particularly in the lower stratosphere. Instrument biases are also influenced by clouds as shown in Jensen et al. (2016) who found systematic differences in temperature measurements greater than 2K between the Vaisala RS92 and RS41 radiosonde when exiting cloud layers. This

large difference in temperature measurements between the two radiosondes was attributed to the wet-bulb effect, where the temperature sensor gets wet while passing through a cloud layer and is subject to evaporative cooling after entering dryer parts of the atmosphere. Below 28km altitude, Jensen et al. (2016) found a mean systematic difference between the temperature measurements of the two radiosondes of 0.13K. For radiosonde measurements performed at GRUAN sites, it is suggested that sites conduct dual sonde launches for at least 6 months when changing from one instrument type to another (GCOS-171,

2013). However, analysis of data from dual sonde launches conducted at the GRUAN Lead Centre suggests that at least 200 dual flights over a period of one year are required to accurately assess the systematic difference between the two sonde-types (GCOS-171, 2013). The number of dual sonde flights required may be site dependent and therefore, site specific analysis is likely required to determine the required number of dual flights at any site. Furthermore, it is possible that instrument biases at one site may not be the same in different atmospheric conditions at other sites, though this has not been extensively evaluated.

Therefore, it would be ideal if all GRUAN sites could complete thorough radiosonde intercomparisons by performing dual




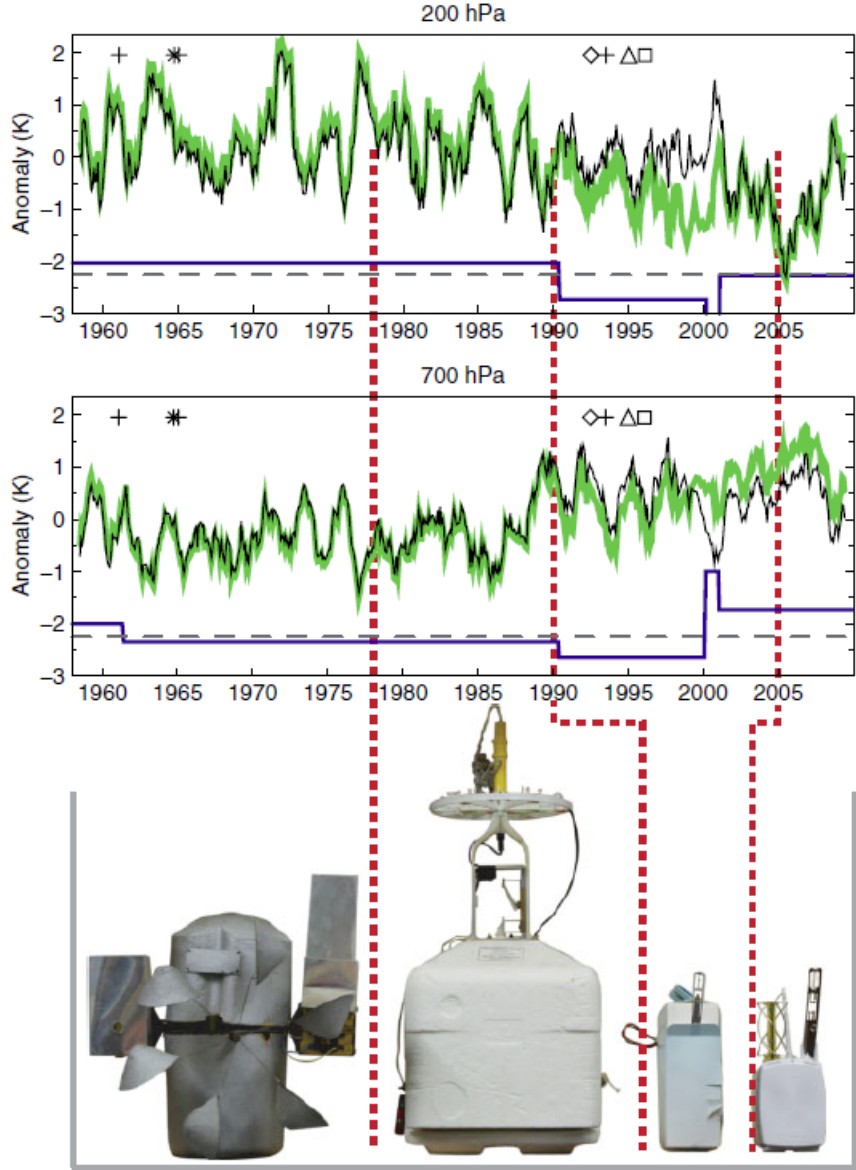

**Figure 1.** Top two panels: Monthly temperature anomalies (smoothed with a 13-point running mean) during 1958-2009 from radiosonde observations at Camborne, Cornwall, UK, at 200 hPa (near tropopause) and 700 hPa (lower-troposphere). Included are raw (black) and adjusted (green) radiosonde temperature data from the Hadley Centre (HadAT). The smoothed difference series between the two shows the adjustments (offset by 2.25 K). Bottom panel: the four radiosonde types used over this period (from left to right, with typical periods of operation): Phillips Mark IIb (1950–1970); Phillips MK3 (mid 1970s to early 1990s); Vaisala RS-80 (early 1990s to 2005–2006); and Vaisala RS-92 (since 2005–2006). Dates of radiosonde changes are indicated by red dotted lines. Five other potential sources of inconsistencies in the data sets include: Change in the radiation correction procedure (cross); Change in the data cut-off (star); Change of pressure sensor (diamond); Change of wind equipment (triangle); Change of relative humidity sensor (square). Figure adapted from Thorne et al. (2011b)





radiosonde launches for at least 6 months prior to any instrument change. However, the costs of such a measurement campaign can be significant, preventing some stations from performing extensive dual launches.

In this study, we investigate the feasibility of quantifying the difference in biases of two instrument types by alternating between the two different instruments and then applying a statistical model to infer any systematic biases between the two in-
struments. For this study, we conduct the investigation by applying the statistical model developed to synthetic data sets, where persistence of weather conditions is a controllable parameter, that represent such interlaced radiosonde flights. Specifically, we investigate (i) if a combination of interlaced measurements together with an appropriate statistical model can be used to estimate the differences in biases of two instrument types, and (ii) if so, how effective the approach is. This method, if feasible, could reduce the financial burden for sites seeking to manage such a transition, since an interlacing approach would not require
additional measurements above what is needed for normal daily operation.

## 2   Methodology

### 2.1   Background

Any modification of instrumentation might introduce a systematic change to the measurement time series. This change is typically assumed to be a constant difference ($\Delta$) as a first order approximation, resulting from differences in the individual
instrument biases, i.e. their systematic deviations from the true value. As the true value of the quantity being measured is unknown in practice, it is not possible to estimate each instrument's individual bias. It is possible, however, to estimate the difference $\Delta = \mathrm{Bias}_A - \mathrm{Bias}_B$ in biases $\mathrm{Bias}_A$ and $\mathrm{Bias}_B$ of instruments $A$ and $B$. If temporally and spatially coincident measurements are made using instrument $A$ and $B$ (i.e. dual flights), this difference can be easily obtained: Consider some quantity of interest, e.g. air temperature ($T$), measured with instrument $A$ and instrument $B$ at the same location and time $t$.
The bias of each instrument is the difference between the expectation value of the instrument's measurement and the unknown true value $T_t$:

$$\mathrm{Bias}(T_{t,A}) = E[T_{t,A}] - T_t \quad \text{and} \quad \mathrm{Bias}(T_{t,B}) = E[T_{t,B}] - T_t. \tag{1}$$

where $T_{t,A}$ and $T_{t,B}$ is the temperature at time $t$ measured with instrument $A$ and $B$, respectively. The difference in the instrumental bias is therefore:

$$\Delta_t = \mathrm{Bias}(T_{t,A}) - \mathrm{Bias}(T_{t,B}) = E[T_{t,A}] - E[T_{t,B}], \tag{2}$$

Consider now that $T_{t,B}$ differs from $T_{t,A}$ only by a constant offset $\Delta$, i.e.:

$$T_{t,A} = T_{t,B} + \Delta \tag{3}$$

which is independent of the true value and thus the measurement time $t$. Under this assumption, an estimate for the stationary difference in biases can be obtained from $N$ dual measurements according to:

$$\hat{\Delta} = \frac{1}{N} \sum_{t=1}^{N} (T_{t,A} - T_{t,B}) = \frac{1}{N} \sum_{t=1}^{N} ((T_{t,A} - T_t) - (T_{t,B} - T_t)), \tag{4}$$





with $\hat{\Delta}$ denoting an estimate of the constant offset $\Delta$. This equation applies even if the true value $T_t$ is changing with time as it depends only on anomalies $T_{t,A/B} - T_t$. Under suitable conditions, the uncertainty (expressed in terms of standard deviation) of this estimate decreases with $\sqrt{N}$ and depends on the persistence (i.e. autocorrelation) of the time series (Wilks, 2011).

## 2.2 A statistical model for interlaced measurements

As dual measurements, using both instrument types, require additional resources, and therefore inherent additional costs, estimating a systematic difference between the instruments using interlaced measurements, i.e. using instrument $A$ at odd days $t \in \{1, 3, 5, \ldots\}$ and instrument $B$ at even days $t \in \{2, 4, 6, \ldots\}$ is explored in this study. Using this approach, at every time $t$ only *one* measurement from *one* instrument is available, hence Eq. 4 is not applicable.

The underlying assumption for the approach outlined here to work is that the quantity of interest fluctuates around a smooth
climatological signal (i.e. a seasonal cycle) and the fluctuations show a certain degree of persistence at the weather time scale, e.g. the fluctuations show a day to day dependence. For a typical difference in the biases between radiosondes this persistence (i.e. autocorrelation) is key to the idea of estimating a bias from interlaced measurements. The difference in the biases tested here is smaller than the day-to-day fluctuations themselves as it carries information from the measurement $A$ to the measurement $B$.

In the following, a simplified model for air temperatures time series complying with the above mentioned assumptions is constructed. The true (unobserved) time series is represented by a smooth seasonal cycle with an auto-regressive process of first order (AR[1], e.g. Box and Jenkins, 1976; Wilks, 2011) added to the time series, i.e.:

$$T_t = \mu_0 + \mu_1 \sin\left(2\pi \frac{d_t}{365} - \frac{\pi}{2}\right) + \mu_2 \sin\left(2\pi \frac{2\,d_t}{365} - \frac{\pi}{2}\right) + \epsilon_t \qquad (5)$$

$$\epsilon_t = a\,\epsilon_{t-1} + \eta_t, \qquad (6)$$

with $d_t \in [1, \ldots, 365]$ giving the day in the year for date $t$, $a$ is the autocorrelation coefficient which describes the degree of persistence in the time series, and $\eta_t \sim \mathcal{N}(0, \sigma^2)$ being the driving noise of the AR[1] process, selected randomly from a Gaussian distribution. The latter is taken to be Gaussian white noise with zero mean and variance $\sigma^2$. This is a well established model for the persistence of, e.g. daily air temperatures (e.g. Wilks, 2011).

Pseudo-observations are now obtained from a realization of $T_t$ (Eq. (5)) with an instrument bias and random measurement
noise added. Here, we aim for interlaced temperature measurements $T_{t,A}$ and $T_{t,B}$ from instruments $A$ and $B$ and thus add the instrument biases $c_A$ and $c_B$, respectively, and independent Gaussian measurement uncertainties $\epsilon_{t,A} \sim \mathcal{N}(0, \sigma_A^2)$ and $\epsilon_{t,B} \sim \mathcal{N}(0, \sigma_B^2)$:

$$T_{t,A} = T_t + c_A + \epsilon_{t,A} \quad t \in t_A = \{1, 3, 5 \ldots\} \quad \text{and} \qquad (7)$$

$$T_{t,B} = T_t + c_B + \epsilon_{t,B} \quad t \in t_B = \{2, 4, 6 \ldots\}. \qquad (8)$$

For simplicity, we assume equal variances $\sigma_A^2 = \sigma_B^2$ for the measurement uncertainties. The continuous series of combined interlaced measurements $T_{t,AB}$ for $t \in \{1, 2, 3, \ldots\}$ is therefore:

$$T_{t,AB} = T_t + c_A\,\chi(t \in t_A) + c_B\,\chi(t \in t_b) + \epsilon_t, \qquad (9)$$



with indicator function $\chi$ being 1 if $t$ is a member of the set $t_A$ or $t_B$, respectively, and 0 otherwise. Figure 2 shows an example of such a synthetic time series of interlaced measurements. This example is based on a simulated temperature time series using a realization of an AR[1] process using an autocorrelation coefficient of $a = 0.5$ in Eq. (6), similar to the autocorrelation coefficient of radiosonde measurements at 300hPa above Lindenberg, Germany (cf. Sec. 2.4).

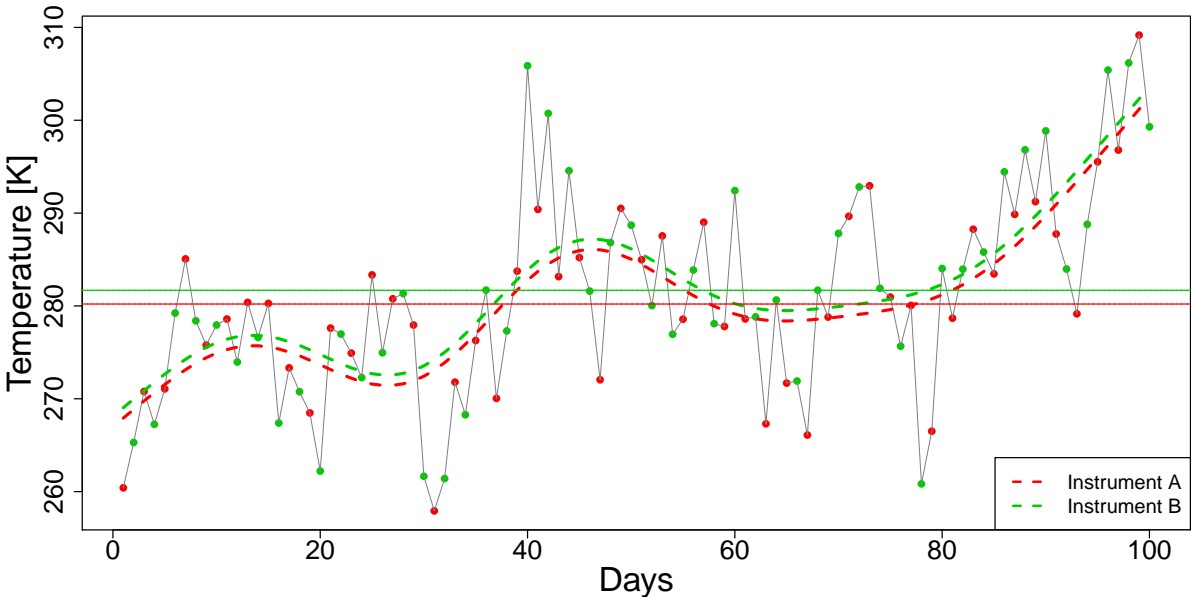

**Figure 2.** Example time series for interlaced measurements of instrument $A$ (red dots) and instrument $B$ (green dots). Horizontal lines are the means of the measurements using instrument $A$ (red) and instrument $B$ (green). Smooth dashed lines (red for instrument $A$, green for instrument $B$) are spline estimates with the differences being an estimate for the differences in the instrument biases.

## 2.3 Estimating the difference in instrument biases

A direct approach to estimate the difference in instrument biases $\Delta = c_A - c_B$ is an estimation using the differences in means $\bar{T}_A$ and $\bar{T}_B$ of instrument $A$ and $B$, respectively, over a common time period $t_1$ to $t_2$, i.e.

$$\widehat{\Delta}_{\mathrm{mean}} = \bar{T}_A - \bar{T}_B \tag{10}$$

with

$$\bar{T}_A = \frac{1}{N_A} \sum_{t \geq t_1}^{t \leq t_2} T_{t,A} \quad \text{for} \quad t \in t_A \quad \text{and} \quad \bar{T}_B = \frac{1}{N_B} \sum_{t \geq t_1}^{t \leq t_2} T_{t,B} \quad \text{for} \quad t \in t_B \tag{11}$$

being the arithmetic means for the individual instruments; $N_A$ and $N_B$ are the number of measurements made by instrument $A$ and $B$, respectively, in the given time period. The uncertainty on this estimate of the difference in instrument biases decreases with increasing $N_A$ and $N_B$ but depends also on the persistence of the underlying time series: larger persistence leads to larger uncertainties when calculating arithmetic means (e.g. von Storch and Zwiers, 1999).





Here, we exploit the persistence and suggest an approach based on the estimation of a slowly varying signal common to both instruments. Imagine, for example, a smooth temperature time series in the absence of weather-induced noise. Measurements are then made of that signal using instrument $A$ and this measurement series is represented by $s(t)$ and an additional measurement noise $\epsilon_t$. Analogously, measurements of the same slowly varying signal are made using instrument $B$ and can be

represented by the same $s(t)$ but with the difference in instrumental biases $\Delta$ and again measurement noise $\epsilon_t$; i.e. $s(t)+\Delta+\epsilon_t$. A model for these interlaced measurements $T_{t,AB}$ is constructed using the indicator function $\chi$:

$$\hat{T}_{t,AB} = s(t) + \Delta\,\chi(t \in t_B) + \epsilon_t\,. \tag{12}$$

For $t \in t_B$, the indicator function $\chi(t \in t_B)$ returns 1 and we obtain a measurement with instrument $B$, i.e. $\hat{T}_{t,B} = s(t)+\Delta+\epsilon_t$. For other time steps $t \in t_A$ the indicator function returns 0 and we obtain a measurement of instrument $A$, i.e. $\hat{T}_{t,A} = s(t)+\epsilon_t$,

excluding the difference in instrumental bias $\Delta$. The statistical model described in Eq. (12) belongs to the class of generalized additive models (GAMs, e.g. Chambers and Hastie, 1992) and the smooth term $s$ can be estimated using a smooth spline fit with degrees of freedom determined by generalized cross validation (Wood, 2006). This functionality is implemented in the R-package `mgcv` (Wood, 2006).

## 2.4   Simulation set-up

To investigate whether interlaced measurements, diagnosed using the methodology described above, can be used to estimate potential biases between instruments, we design a simulation study wherein an ensemble of synthetic upper-air temperature time series is generated using a stochastic process. For each member of the ensemble, interlaced measurements for two instruments are obtained by adding a systematic measurement uncertainty (i.e. bias) for each instrument plus some random measurement noise. As the instrument biases are known, their difference $\Delta$ is also known. The questions to be answered in

this study are:

1. Can a combination of interlaced measurements, together with an adequate statistical model, be used to estimate the difference in instrument biases?

2. If so, how effective is this estimation compared to an approach requiring dual measurements?

An analysis of the 300hPa temperatures measured by radiosondes at Lindenberg, Germany, forms the basis for this simulation

study. After subtracting the seasonal cycle, the temperature anomalies show a variance of about $\sigma^2_{\text{anomalies}} = 10\text{K}^2$ and can be adequately described with an AR[1] process as in Eq. (6) with $a \sim 0.5$. To provide a realistic synthetic time series for analysis, we use driving Gaussian white noise $\eta \sim \mathcal{N}(0,\sigma_a^2)$ with variance $\sigma_a^2 = (1 - a^2)\,\sigma^2_{\text{anomalies}}$. This choice of $\sigma_a^2$ ensures that the anomaly variance is fixed at $\sigma^2_{\text{anomalies}} = 10\text{K}^2$ independent of the value of $a$. This is necessary as we vary the persistence parameter (i.e. the autocorrelation coefficient) $a \in (0,1)$ to study time series with different persistence but identical anomaly

variance.

The synthetic temperature series is generated using Eq. (9) that includes a seasonal cycle and a realization of an AR[1] process. The instrument biases in Eq. (9), are prescribed at $c_A = -0.1\text{K}$ and $c_B = 0.2\text{K}$ and are added to the time series to-





gether with a measurement uncertainty being specified as Gaussian white noise $\epsilon \sim \mathcal{N}(0, \sigma^2)$. The resulting two time series for instruments $A$ and $B$ are combined to a) a synthetic time series of dual measurements, and b) an interlaced observational counterpart. The difference in instrument biases between both time series is prescribed as $\Delta = c_A - c_B = -0.1 - 0.2 = -0.3$K. To investigate the influence of (i) persistence in the temperature series, (ii) measurement noise, and (iii) number of measure-

ments on our ability to estimate the difference in biases between two instruments, the following parameters are prescribed and controlled in our study:

**persistence of the time series** $a \in \{0.5, 0.7, 0.8, 0.9, 0.95, 0.99\}$

**number of measurements** $N \in \{50, 100, 250, 500, 1000, 2000, 3000\}$,

leading to $6 \times 7 = 42$ combinations, i.e. 42 synthetic time series to be analysed. The instrument noise is fixed at $\sigma^2 \in 0.1$. To

generate a synthetic time series for a given $a$, $N$ and $\sigma$, the following steps were taken:

1. Generate a time series of length $N$ consisting of an annual cycle and a realization of an AR[1] process as described above.

2. Add an offset of -0.1K (instrument bias of instrument $A$) and Gaussian noise with variance $\sigma^2 = 0.1$ to produce a synthetic time series for instrument $A$.

3. Add an offset of $0.2$K (instrument bias of instrument $B$) and Gaussian noise with variance $\sigma^2 = 0.1$ to produce a synthetic time series for instrument $B$.

4. Select measurements from $A$ for odd days and from $B$ for even days to generate an interlaced time series.

5. Repeat steps 1 to 4 many times (e.g. $M = 1000$) to generate 1000 synthetic time series to derive statistically robust estimates of $\hat{\Delta}$.

The difference in instrument biases is then estimated based on

1. the calculated mean values of $N$ dual measurements (Eq. (10)), i.e. $N$ measurements for $A$ and $N$ measurements for $B$ made simultaneously, and

2. results from the statistical model (Eq. (12) using the time series of $N$ interlaced measurement, i.e. $N/2$ measurements for $A$ and $N/2$ measurements for $B$.

## 3 Results

The box plots in Fig. 3 summarize the distribution of $M = 1000$ bias estimates $\hat{\Delta}$ for a varying numbers of interlaced flights $N$. The upper panel of Fig. 3 is based on the simulated temperature time series with an AR[1] coefficient $a = 0.5$, being similar to the autocorrelation coefficient found for temperature measurements at 300hPa above Lindenberg. The middle and bottom rows



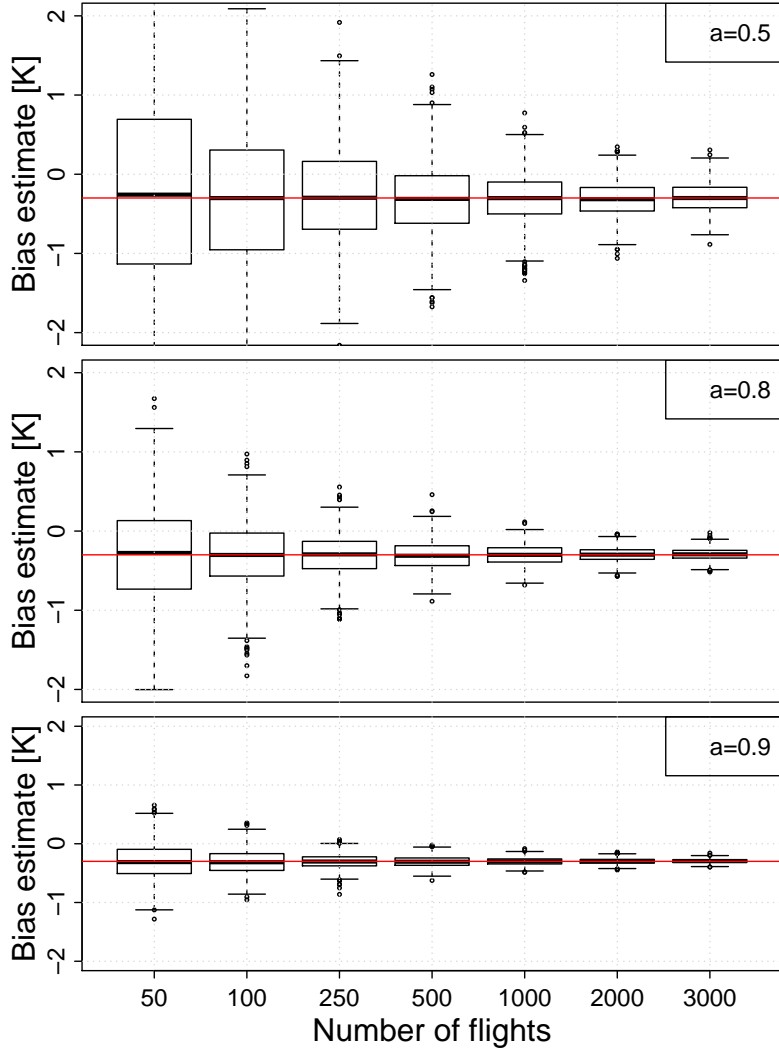

**Figure 3.** Box and whisker plots of bias estimates ($\widehat{\Delta}$) against number of interlaced flights $N$ (50 flights means 25 flights of instrument $A$ and 25 flights of instrument $B$) as derived from $M = 1000$ simulations using an autocorrelation coefficient of $a = 0.5$ (top), $a = 0.8$ (middle) and $a = 0.9$ (bottom) and a measurement noise of $\sigma^2 = 0.1$. The boxes show the inter-quartile range. The upper and lower whiskers represent the maximum (excluding outliers) and minimum (excluding outliers). Suspected outliers are shown as dots and are located outside the fences ("whiskers") of the boxplot (e.g: outside 1.5 times the interquartile range above the upper quartile and below the lower quartile). The true difference in biases $\Delta = -0.3$K is marked with a red line.





are examples for stronger persistence, i.e. $a = 0.8$ and $a = 0.9$, respectively. All panels show that the spread in the estimated difference in bias between instruments $A$ and $B$ ($\widehat{\Delta}$) converges towards the true value ($\Delta = -0.3$) for increasing $N$ in all cases. The rate at which this converges with increasing $N$ depends on the persistence (i.e. autocorrelation) in the underlying time series. Weak persistence (small $a$) leads to slower convergence (Fig. 3, top row), while strong persistence ($a$ approaching

1) shows faster convergence.

The standard deviation of $\widehat{\Delta}$ (see Fig. 4), representing the uncertainty with which the difference in the bias between instruments $A$ and $B$ can be estimated, depends on the number of interlaced flights and on the AR[1] coefficient $a$ (coloured lines in Fig. 4). The standard deviation can be used to construct asymptotic confidence intervals for the estimates using the standard normal assumption (e.g. Wilks, 2011, Chapt. 5), i.e. for a 95% confidence interval, the estimated bias needs to be within 1.96

times the standard deviation. For all $a$, the standard deviation decreases with increasing $N$; however, the standard deviation is generally larger for weak persistence (small $a \in (0,1)$) and smaller for strong persistent (large $a \in (0,1)$).

The synthetic time series of dual flights performed with instrument $A$ and $B$ simultaneously at N times (i.e. $2N$ measurements, solid black line in Fig. 4) provides the most reliable estimate of the biases between the instruments, i.e. the standard deviation is smallest for any $N$. To provide a robust comparison of the results from the dual flights to the results from $N$ inter-

laced measurements, the results from the dual flights need to be compared to the results of doubled $N$ interlaced flights. For a time series with an autocorrelation coefficient of $a = 0.5$, at least 2000 days of consecutive interlaced daily measurements would be required to estimate the difference in instrument's biases with a standard deviation of 0.22 K. Consider the following example, a station operator seeks to detect the difference in bias between two radiosondes in a temperature time series showing an autocorrelation coefficient of 0.95. The station operator requires a standard deviation of $\hat{\Delta} \leq 0.05\text{K}$ which leads to a 95%

confidence interval of about 0.1 K ($\approx 0.05 * 1.96$), then, from Fig. 4 it can be inferred that 500 interlaced measurements are required to achieve this. Furthermore we conclude that, if an operator has a given amount of two types of radiosondes available from which the difference in instrument biases needs to be estimated, it is clear from Fig. 4 that dual flights result in better estimates (i.e. smaller standard deviation in Fig. 4) than to interlace the instrument types from one day to the next. The results presented here (from dual and interlaced flights) also depend on the variance of the signal; for a higher measurement noise, the

number of required days will increase and vice versa (not shown).

The results indicate that for typical difference in biases between radiosonde types, the presented method on interlaced measurements is unlikely to provide a robust estimate of the difference in biases for a reasonable length of the measurement period (reasonable is considered as 2 years here). That said, there might be cases of larger instrument biases and/or larger persistence where the interlaced method could provide an alternative method to dual measurements, requiring fewer resources.

This, however, is outside the scope of this study which focuses on radiosonde temperature measurements.

## 4  Conclusions

We have used synthetic time series representing temperature measurements to investigate the possibility of using interlaced measurements performed with two different instruments types together with Generalized Additive Models to obtain an estimate



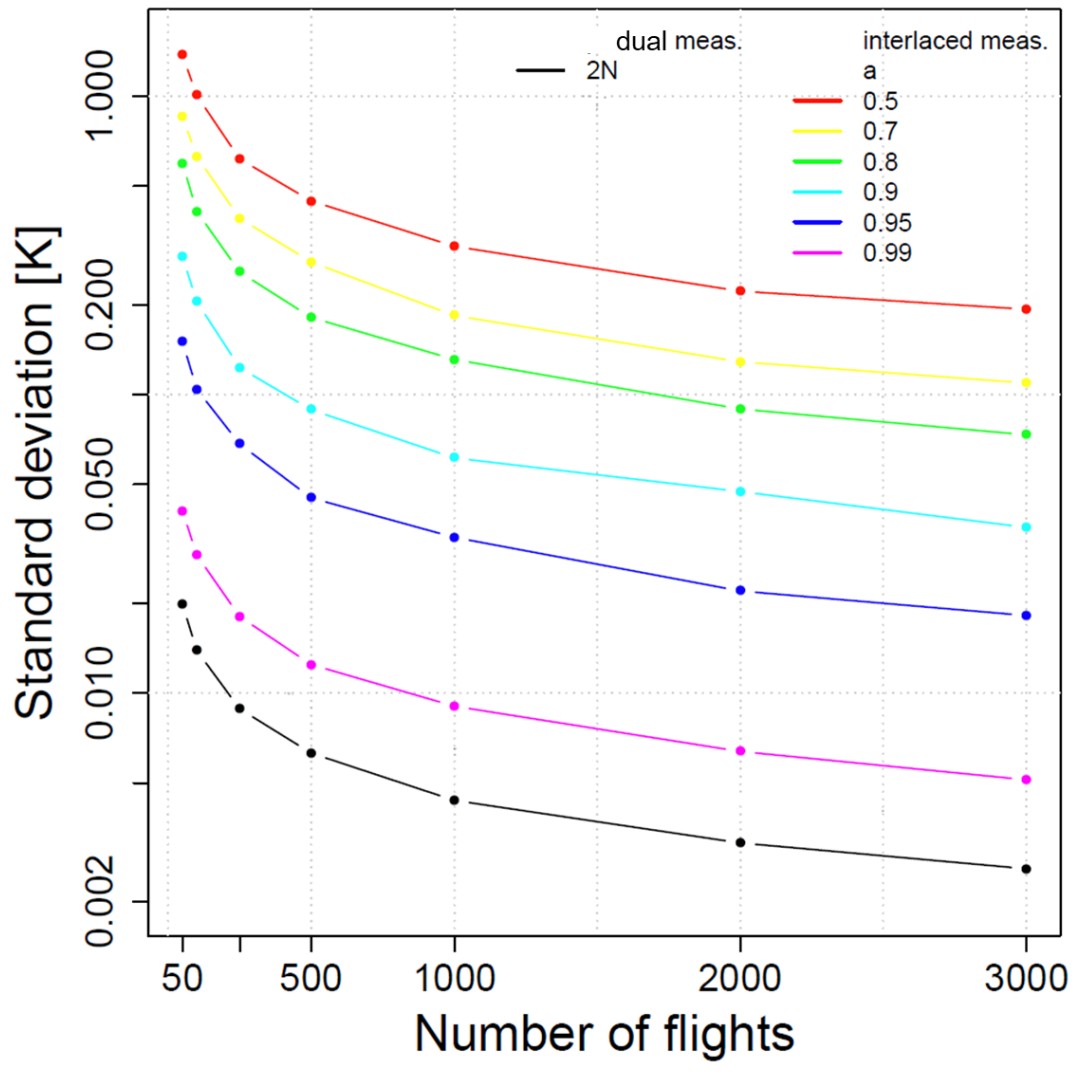

**Figure 4.** Standard deviation of $\widehat{\Delta}$ against number of flights $N$ for different AR[1] coefficients $a$. The black solid line represents the reference experiment with dual flights of instruments $A$ and $B$, i.e. $2N$ measurements. To compare the results from the dual flights (black solid line) with the results obtained from interlaced flights, the number of dual flights have to be doubled. Note the logarithmic vertical scale.



of the difference in the bias between the two instrument types. Performing dual radiosonde flights with both instrument types is costly, and therefore we investigated the feasibility of using interlaced flights to obtain an estimate of the difference in the bias. This would be more sustainable and less costly. Information about typically small differences in instrument biases can be obtained from non-simultaneous measurements using a persistence assumption, i.e. some information of some day's

measurement is carried over to the next day. As atmospheric temperatures tend to be autocorrelated in time (e.g. Wilks, 2011; Maraun et al., 2004), the persistence assumption is justifiable. However, the strength of the autocorrelation depends, in part, on the geographical location of the measurement site and on altitude. Here we investigated how a statistical approach to estimate the difference between two instrument biases is affected by the persistence of a time series.

The results presented here indicate that while it is in principle possible to estimate the difference between two instrument's

biases from interlaced measurements, the number of interlaced flights required to obtain a satisfying accuracy is very large for reasonable values of the autocorrelation coefficient. Strongly autocorrelated signals require fewer data for an accurate estimate of the difference in biases and therefore fewer interlaced flights, than time series with low autocorrelation. The results show that for very strong persistence (e.g. an AR[1] coefficient of 0.99) about twice the number of measurements is needed compared to parallel measurements to obtain a comparable uncertainty in estimates for interlaced measurements. Hence, the described

approach may be used for measurements with very strong persistence or where the costs for sufficient parallel measurements exceeds the costs for sufficient interlaced measurements to confidently infer the difference in the instrument bias. However, if, for example, it would be possible to derive a robust estimate of the difference in instrument biases from interlaced measurements in some reasonable time period (e.g. 2 years) and even if this period was more than 2 or 3 times longer than would be required from a dual measurement strategy to achieve the same level of confidence, the interlacing approach would present a case saving

over an approach that would start with dual flights and then continue with flights using only the new instrument.

*Code availability.* The code can be obtained by contacting the corresponding author.

*Competing interests.* The authors declare that they have no conflict of interest

*Acknowledgements.* We would like to thank the NOAA GCOS office, through the Meteorological Service of New Zealand Limited, for supporting this research. HR acknowledges support from the Freie Universität Berlin within the Excellence Initiative of the German Research

Foundation. We would also like to thank Fabio Madonna and Alessandro Fasso for helpful discussion around the alternative approach of interlaced measurements. We thank the GCOS Reference Upper-Air Network (GRUAN) for providing the data used in this publication. The data were sourced from ftp://ftp.ncdc.noaa.gov/pub/data/gruan/processing/level2/. The authors confirm that these data have been used in a manner consistent with the GRUAN data use policy, as articulated in the GRUAN Guide, and have not been used for commercial gain.




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
