# Peer review of "Is it feasible to estimate radiosonde biases from interlaced measurements?"

_Atmospheric Measurement Techniques, 2018_

## Referee Comment (RC1) · Anonymous Referee #1 · 22 Feb 2018

When radiosonde instruments change it is essential to estimate any systematic differences between measurements from the old and new instruments as accurately as possible. Such estimates can be most accurately obtained from ascents with both instruments attached on the same balloon over a certain period of time (order of 1 year) or from interlaced measurements, thereby avoiding the extra cost of ttwo measurements per flight.

The authors show convincingly that the interlaced approach yields much less accurate estimates, particularly if autocorrelation is low. In order to achieve a target accuracy of 0.1 K at least 2 years of interlaced measurements (if launched once per day) are needed for autocorrelation coefficients between 0.9 and 0.95. This may be longer than the lifetime of a particular radiosonde model. And also this yields only an annual mean

bias estimate, no bias estimates as a function of e.g. solar elevation.

The paper convincingly makes the case for the need of dual measurements to achieve the quality of bias estimates needed in a reference network such as GRUAN.

What I am missing as a reader are one or two vertical profiles of autocorrelation coefficients calculated from radiosonde temperature time series in the Tropics and in the Extratropics. This would be helpful for estimating which curve in Fig. 4 is the most relevant one. So far the paper only states that the autocorrelation at Lindenberg is around 0.5. The autocorrelation at other levels and regions may be quite different.

Apart from that minor issue the paper is well structured and well written and I recomment publication with the suggested minor revision.

---

## Referee Comment (RC2) · Anonymous Referee #2 · 20 Mar 2018

This paper investigates potential usefulness of "interlaced" comparison measurements from two different radiosonde types, rather than "dual" measurements with single balloons, being motivated to reduce the cost that the sites need to cover for dual measurements when changing from one instrument type to another. The authors prepare several different sets of simulated dual/interlaced measurement data with specified instrument biases, different autocorrelation coefficients (a measure of high frequency weather disturbances), and different numbers of flights, and calculate the standard deviation (a measure of uncertainty) of estimated instrument biases. The authors find that the standard deviation reduces as the number of flights increases or as the autocorrelation coefficient increases (i.e., weather variability decreases), but virtually never reaches the level that dual measurements would provide. Thus, they conclude that the

method on interlaced comparison measurements is "unlikely to provide a robust esti-
mate of the difference in biases for a reasonable length of the measurement period"
(e.g., 2 years).

This is an important contribution to the climate science community by providing a
statistics-based evaluation of usefulness of the method of interlaced measurements
that some operational sites or researchers may consider as an option. The methodol-
ogy of the paper is sound and basically explained well. The manuscript is appropriate
for AMT, and should be published once the following minor comments and suggestions
are considered.

Specific comments.

Page 2, line 21: Kobayashi et al. (2012) also give a very good example of dual sounding
program (a total of 115 dual soundings for four different seasons) at a GRUAN site,
Tateno when they changed from Meisei RS2-91 to Vaisala RS92.

Kobayashi, E., Y. Noto, S. Wakino, H. Yoshii, T. Ohyoshi, S. Saito, and Y. Baba, 2012:
Comparison of Meisei RS2-91 rawinsondes and Vaisala RS92-SGP radiosondes at
Tateno for the data continuity for climatic data analysis. J. Meteor. Soc. Japan, 90,
923−945, https://doi.org/10.2151/jmsj.2012-605.

Page 2, around line 21: WMO conducted several radiosonde intercomparison cam-
paigns in the past (e.g., Nash et al., 2011; Jeannet et al., 2008 and the references
therein). It would be fair to mention these and discuss its usefulness and/or limitations.

Nash, J., T. Oakley, H. Vömel, and L. Wei, 2011: WMO intercomparison of high quality
radiosonde systems, Yangjiang, China, 12 July-3 August 2010. WMO/TD No.1580.
IOM Report, No.107, World Meteorological Organization, Geneva, 248pp., available at
https://library.wmo.int/opac/index.php?lvl=notice_display&id=15531#.WrBryGrFJ0w

Jeannet, P., C. Bower, B. Calpini, 2008: Global criteria for tracing the improve-
ments of radiosondes over the last decades, WMO/TD No. 1433, IOM Report No. 95, World Meteorological Organization, Geneva, 32 pp., available at https://library.wmo.int/opac/index.php?lvl=notice_display&id=15522#.WrBt_WrFJ0w

Figure 1, caption. Please add the explanation on the dotted and blue lines in the upper two panels.

Page 4, Equation 4: Why the delta-hat is not E[ delta ]? A sentence explaining the reason for this at line 28 may be useful for readers.

Page 5, Equation 5: Why there is a phase component "-pi/2"? Also, in general, there should be cosine components as well for both diurnal and semi-diurnal variations? If, for this simulation study, it is enough to consider sine components only, mention that perhaps at line 23.

Page 5, lines 20-21: The key word "weather" has already appeared at line 10, but it would be useful to mention it again when "a" first appears here, so that the readers are reminded that "a" is the one related to the magnitude of high frequency weather variability which is "noise" in this study. Something like: "(or the magnitude of weather-related variability, larger for smaller a)"

Page 6, line 13-14: "larger persistence lead to larger uncertainties" – isn't it possible to show an equation for this using "a"?

Page 7, lines 10-13: It would be nice to have some more explanation on the GAMs. (Are the GAMs a class of statistical distributions that Tt,AB would follow? What factors/components determine the degrees of freedom here?) This is in part because the authors mention the GAMs again at the second line of the Conclusions, as a key component for this study.

Page 7, line 30: I assume that 300 hPa at Lindenberg (a midlatitude site) would give near-largest weather-related variability, i.e., minimum "a", compared to other height regions and other latitude regions. But, I think it would be useful to actually show this by showing the values of "a" for other height regions at Lindenberg (and perhaps at a

tropical site as well).

Page 8, lines 9-10: Please also add explanation on M here.

Page 10, line 25, and lines 28-29: Showing a figure on this might be useful? Also, stratospheric water vapor measurements may be an example for this?

Page 12, Competing interests: The period is missing at the end of the sentence.
* * *

---

## Author Comment (AC1) · 24 Apr 2018

We appreciate the suggestions and constructive comments provided by both reviewers. Below, the reviewer's comment is repeated in blue with our response in black.

**Response to Reviewer 1**

What I am missing as a reader are one or two vertical profiles of autocorrelation coefficients calculated from radiosonde temperature time series in the Tropics and in the Extratropics. This would be helpful for estimating which curve in Fig. 4 is the most relevant one. So far the paper only states that the autocorrelation at Lindenberg is around 0.5. The autocorrelation at other levels and regions may be quite different.

[Figure]

We thank the reviewer for this suggestion and we have now added a new figure to the manuscript (Figure 5). This figure shows vertical profiles of autocorrelation coefficients determined from ERA5 reanalyses interpolated to the locations of 6 GRUAN sites, including sites in the tropics, middle and high latitudes. We chose to calculate the autocorrelation coefficients from ERA5 data rather than from radiosondes as long-term continuous measurements are required to obtain a robust estimate of the seasonal cycle of the temperature time series before calculating the autocorrelation coefficients. Such continuous observations, covering at least 2 years of daily radiosonde flights, are currently only available at a small subset of GRUAN sites, which does not cover all latitude bands. ERA5 is the latest reanalysis data set provided by ECMWF and it is expected that the calculated autocorrelation coefficients provide a good estimate of the autocorrelation coefficient at each of the selected sites. The estimated autocorrelation coefficient at 300 hPa for the radiosonde measurements made at Lindenberg (0.5 as described in the manuscript), agrees very well with the coefficient determined from the ERA5 reanalyses.

---

## Author Comment (AC2) · 24 Apr 2018

We appreciate the suggestions and constructive comments provided by both reviewers. Below, the reviewer's comment is repeated in bold with our response in black.

**Response to Reviewer 2**

Page 2, line 21: Kobayashi et al. (2012) also give a very good example of dual sounding program (a total of 115 dual soundings for four different seasons) at a GRUAN site, Tateno when they changed from Meisei RS2-91 to Vaisala RS92.

We added the reference to the revised manuscript as suggested by the reviewer.

[Figure]

Page 2, around line 21: WMO conducted several radiosonde intercomparison campaigns in the past (e.g., Nash et al., 2011; Jeannet et al., 2008 and the references therein). It would be fair to mention these and discuss its usefulness and/or limitations.

We included both references and the following sentences in the revised manuscript: "In the past, WMO conducted several radiosonde intercomparison campaigns (e.g. Jeannet et al. 2008 and Nash et al. 2011) with the objective of investigating the performance of operational radiosonde systems. The results of these campaigns are used, in part, to improve the accuracy of daytime operational radiosonde measurements and the associated correction procedures to provide temperature and relative humidity accuracies currently possible with night time measurements. The knowledge of the performance that can be expected from various radiosonde systems allows the users to make a well informed decision on the choice of future equipment. For a measurement network like GRUAN, it is essential to have more than one good quality radiosonde type for operations."

Figure 1, caption. Please add the explanation on the dotted and blue lines in the upper two panels.

We have clarified what the dashed and blue lines in Figure 1 represent.

Page 4, Equation 4: Why the delta-hat is not E[ delta ]? A sentence explaining the reason for this at line 28 may be useful for readers.

E[delta] is the expectation value (sometimes called 'true' value) of the constant offset delta, conceived as a random variable. $delta_{hat}$ is its estimator, used to obtain an estimate for the unknown true value from observations.

Page 5, Equation 5: Why there is a phase component "-pi/2"? Also, in general, there should be cosine components as well for both diurnal and semi-diurnal variations? If, for this simulation study, it is enough to consider sine components only, mention that perhaps at line 23.

Combining sine and cosine of the frequency w is equivalent to using sine OR cosine with a phase shift phi, e.g. a*sin(w) + b*cos(w) = A*sin(w+phi) with A=sqrt(a$^2$ +b$^2$) and phi=atan2(b/a), see also the text book of Daniel Wilks Statistical Methods for the Atmospheric Sciences (2010), Chap. 8.4.3

Page 5, lines 20-21: The key word "weather" has already appeared at line 10, but it would be useful to mention it again when "a" first appears here, so that the readers are reminded that "a" is the one related to the magnitude of high frequency weather variability which is "noise" in this study. Something like: "(or the magnitude of weather related variability, larger for smaller a)"

We followed the suggestion by the reviewer and added: '$a$ is the autocorrelation coefficient which describes the degree of persistence in the time series at the weather time scale, e.g. the fluctuations show a day to day dependence, ...'

Page 6, line 13-14: "larger persistence lead to larger uncertainties" – isn't it possible to show an equation for this using "a"?

Such an equation is given in the text book of von Storch and Zwiers (1999), Chap. 17, which we include as a reference. At this point we do not see it to be useful to discuss this basic issue here. It refers to 'arithmetic mean' calculations as stated in the paper.

Page 7, lines 10-13: It would be nice to have some more explanation on the

[Figure]

GAMs. (Are the GAMs a class of statistical distributions that Tt,AB would follow? What factors/ components determine the degrees of freedom here?) This is in part because the authors mention the GAMs again at the second line of the Conclusions, as a key component for this study.

Generalized additive models are a fundamental class of regression models which, other than generalized linear models, allow for nonlinear but smooth components - such as splines. A very good introduction is given in the text book of Simon Woods which we cited. We changed the text in Sect 2.3 to:
'The statistical model described in Eq. (12) belongs to the class of generalized additive models (GAMs, e.g. Chambers and Hastie, 1992), a fundamental class of regression models. GAMs extend generalized linear models (or "linear regression) by introducing additionally to the classical linear components a smooth term s. This term can be estimated using a smooth spline fit with its degrees of freedom (its flexibility of smoothness) determined by generalized cross validation (Wood, 2006).'

Page 7, line 30: I assume that 300 hPa at Lindenberg (a midlatitude site) would give near-largest weather-related variability, i.e., minimum "a", compared to other height regions and other latitude regions. But, I think it would be useful to actually show this by showing the values of "a" for other height regions at Lindenberg (and perhaps at a tropical site as well).

We agree with the reviewer and we have now added a new figure to the manuscript (Figure 5). This figure shows vertical profiles of autocorrelation coefficients determined from ERA5 reanalyses interpolated to the locations of 6 GRUAN sites, including sites in the tropics, middle and high latitudes. We chose to calculate the autocorrelation coefficients from ERA5 data rather than from radiosondes as long-term continuous measurements are required to obtain a robust estimate of the seasonal cycle of the temperature time series before calculating the autocorrelation coefficients. Such

continuous observations, covering at least 2 years of daily radiosonde flights, are currently only available at a small subset of GRUAN sites, which does not cover all latitude bands. ERA5 is the latest reanalysis data set provided by ECMWF and it is expected that the calculated autocorrelation coefficients provide a good estimate of the autocorrelation coefficient at each of the selected sites. The estimated autocorrelation coefficient at 300 hPa for the radiosonde measurements made at Lindenberg (0.5 as described in the manuscript), agrees very well with the coefficient determined from the ERA5 reanalyses.

Page 8, lines 9-10: Please also add explanation on M here.

We assumed that the reviewer referred to line 18 and included an explanation for M in the revised manuscript.

Page 10, line 25, and lines 28-29: Showing a figure on this might be useful?

While we agree with the reviewer that an additional figure might be useful, we decided not to perform additional calculations for other synthetic time series with increased measurement noise as the focus of this paper is on describing the method for determining the differences in the instrument bias, and an in-depth analysis of the applicability of this method for different variances or persistence is considered to be beyond the scope of this short paper. As the software used in this study can be obtained from the authors, the calculations can be repeated by others for their specific measurement time series as variance and persistence vary from site to site.

Also, stratospheric water vapor measurements may be an example for this?

We agree with the reviewer that stratospheric water vapour might have a higher persistence than temperature and it could be tested whether or not the described

interlacing approached is applicable to deriving differences in biases in measurements obtained from, e.g. frost point hygrometers. Radiosondes measurements of stratospheric water vapour, however, are highly uncertain and have limited value in this context. Therefore, we have not discussed the applicability of this interlacing method to water vapour measurements in this paper, which focuses on radiosonde temperature measurements.

Page 12, Competing interests: The period is missing at the end of the sentence.

Done.
* * *